# Structural insights into CED-3 activation

Yini Li[1,*] , Lu Tian[1,*] , Ying Zhang[1] , Yigong Shi[1,2,3,4]

In *Caenorhabditis elegans (C. elegans)*, onset of programmed cell death is marked with the activation of CED-3, a process that requires assembly of the CED-4 apoptosome. Activated CED-3 forms a holoenzyme with the CED-4 apoptosome to cleave a wide range of substrates, leading to irreversible cell death. Despite decades of investigations, the underlying mechanism of CED-4–facilitated CED-3 activation remains elusive. Here, we report cryo-EM structures of the CED-4 apoptosome and three distinct CED-4/CED-3 complexes that mimic different activation stages for CED-3. In addition to the previously reported octamer in crystal structures, CED-4, alone or in complex with CED-3, exists in multiple oligomeric states. Supported by biochemical analyses, we show that the conserved CARD–CARD interaction promotes CED-3 activation, and initiation of programmed cell death is regulated by the dynamic organization of the CED-4 apoptosome.

## Introduction

Programmed cell death (PCD), also known as apoptosis, is conserved in metazoans and plays essential roles in the development of multicellular organisms and maintenance of homeostasis (Horvitz, 2003; Danial & Korsmeyer, 2004). Execution of apoptosis is characterized by the activation of cell-killing proteases known as caspases, which comprise initiator and effector caspases. Initiator caspases are activated by a protein machinery such as the apoptosome, whereas effector caspases can be directly cleaved and activated by initiator caspases (Yan & Shi, 2005). The conserved apoptotic pathway was first identified in *Caenorhabditis elegans* (Horvitz et al, 1994), in which CED-3 (cell death abnormal, CED) is the only caspase (Yuan et al, 1993; Xue et al, 1996).

CED-3 is synthesized as an inactive zymogen; its activation requires association with the CED-4 apoptosome (Yuan & Horvitz, 1992; Chinnaiyan et al, 1997; Irmler et al, 1997; Seshagiri & Miller, 1997; Wu et al, 1997; Yang et al, 1998). Under non-apoptotic condition, dimeric CED-4 is sequestered by the Bcl-2 family member CED-9 (Hengartner & Horvitz, 1994; Chinnaiyan et al, 1997; Spector et al, 1997; Yan et al, 2005) (Fig 1A). During development, 131 cells in a *C. elegans* are programed to die at specific time and specific location, resulting in 959 cells in the adult worm. To initiate cell death, EGL-1 is transcriptionally activated and specifically binds to CED-9 (Conradt & Horvitz, 1998; del Peso et al, 1998; Yan et al, 2004). Association of EGL-1 with CED-9 induces a conformational change of the latter, making it incompatible with CED-4 interaction (Yan et al, 2004, 2005). The released CED-4 dimer oligomerizes to form a functional apoptosome. Crystal structure of the octameric CED-4 apoptosome, arranged as a tetramer of asymmetric dimers, reveals a funnel-shaped architecture, in which eight caspase recruitment domains (CARDs) form two layers of tetrameric rings on the narrow end and the nucleotide-binding oligomerization domain (NOD) enclose a larger ring (Qi et al, 2010; Huang et al, 2013).

In the presence of the CED-4 apoptosome, CED-3 undergoes autocleavages at multiple sites, resulting in separation of the CARD domain and the catalytic domain, the latter comprising the large and small subunits (Hugunin et al, 1996; Seshagiri & Miller, 1997). Preliminary cryo-EM analysis indicates that the catalytic domain of CED-3 is accommodated in the hutch of the CED-4 apoptosome (Qi et al, 2010). Crystal structure of the L2' loop (residues 389–406) peptide of CED-3 bound to the CED-4 apoptosome unveils specific interactions between the CED-3 catalytic domain and CED-4 (Huang et al, 2013). However, the role of the conserved CARD–CARD interaction between CED-3 and CED-4 remains unclear.

To achieve an advanced understanding of the activation mechanism of CED-3 by CED-4, we assembled various CED-4/CED-3 complexes to mimic the sequential activation stages of CED-3 and determined their cryo-EM structures. The CARD–CARD interaction between CED-4 and CED-3 is thoroughly characterized within the structure of the holoenzyme. Unexpectedly, we observe multiple oligomeric states of the CED-4 apoptosome and the CED-4/CED-3 complexes, which afford unprecedented mechanistic insights into CED-4 facilitated CED-3 activation.

## Results

### Cryo-EM structural determination of the CED-4 apoptosome

In the structure of the CED-4 apoptosome co-crystallized with the catalytic domain of CED-3, no density was found for CED-3,

[1]Beijing Frontier Research Center for Biological Structures, Tsinghua-Peking Center for Life Sciences, School of Life Sciences, Tsinghua University, Beijing, China  [2]Westlake Laboratory of Life Science and Biomedicine, Westlake Institute for Advanced Study, Hangzhou, China  [3]Key Laboratory of Structural Biology of Zhejiang Province, School of Life Sciences, Westlake University, Hangzhou, China  [4]Institute of Biology, Westlake Institute for Advanced Study, Hangzhou, China

Correspondence: liyini@mail.tsinghua.edu.cn; shi-lab@tsinghua.edu.cn
*Yini Li and Lu Tian contributed equally to this work

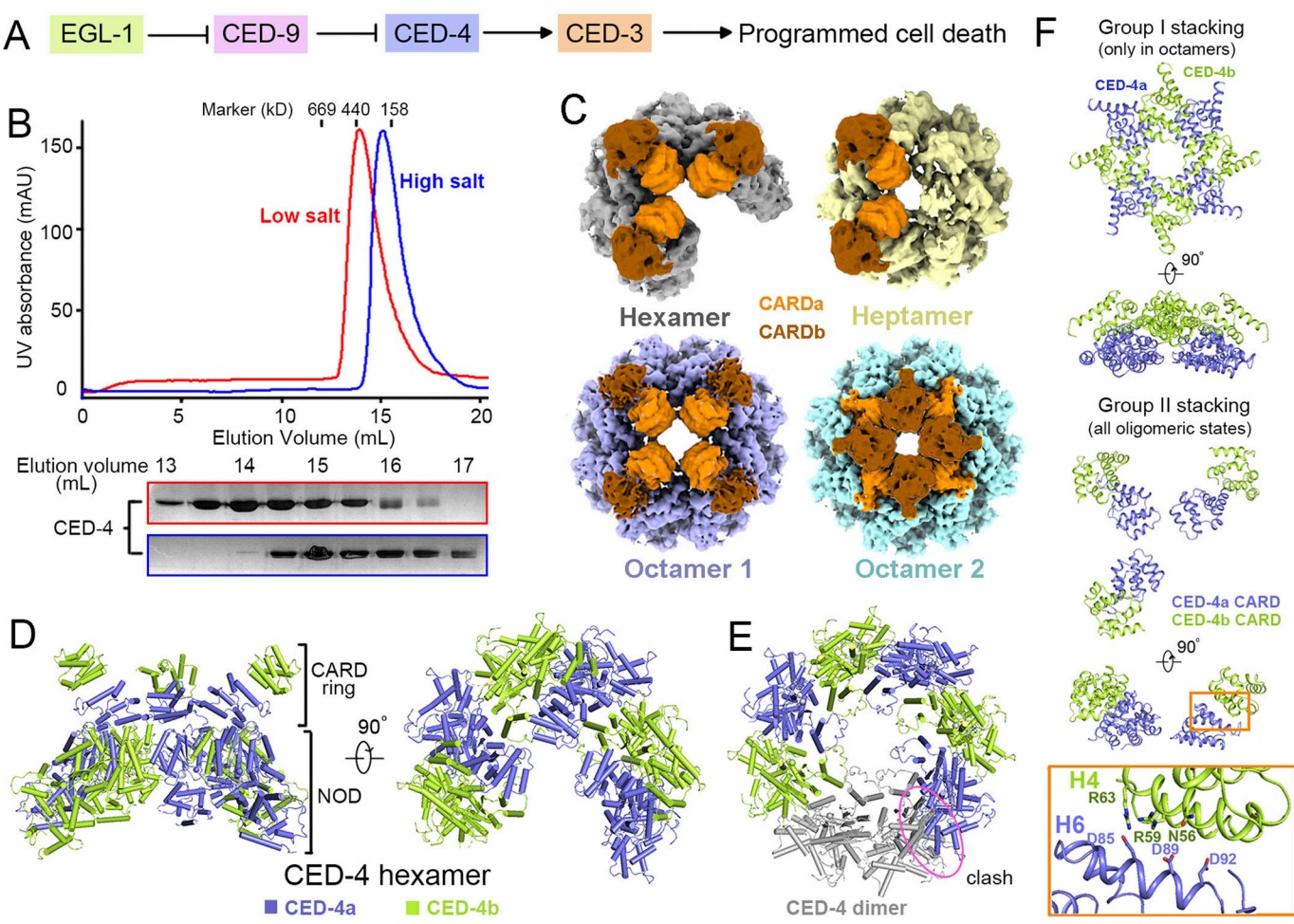

**Figure 1. Cryo-EM structures of the CED-4 apoptosome.**
**(A)** Four proteins—EGL-1, CED-9, CED-4, CED-3—act in a linear fashion to control the onset of PCD in *C. elegans*. **(B)** Different oligomerization states of the CED-4 apoptosome in high- and low-salt buffers. Shown here are representative chromatograms of the CED-4 apoptosome eluted from SEC pre-equilibrated in high- (blue) or low-(red) salt buffers. Peak fractions were loaded to SDS–PAGE and visualized via Coomassie blue staining. **(C)** Four cryo-EM maps of the CED-4 apoptosome, including a hexamer (grey), a heptamer (yellow), and two octamers (cyan and slate) were obtained. CED-4 CARDs are colored in orange and brown to indicate different layers. **(D)** Two perpendicular views of the overall structure of the hexameric CED-4 apoptosome. The two protomers CED-4a and CED-4b are colored slate and limon, respectively. **(E)** The CED-4 hexamer loses C4 symmetry. The hexamer cannot accommodate a fourth CED-4 dimer (grey). Potential steric clash with a fourth CED-4 dimer is labeled by the magenta oval. **(F)** CED-4 CARDs have two stacking modes in different oligomers. Group I stacking only exists in octamer 2. Group II stacking, which has not been observed previously, is found in three oligomeric states. Group II stacking is likely to be maintained by salt bridges between helices H6 and H4 (orange rectangle).

presumably owing to signal averaging as a result of symmetry operation (Qi et al, 2010). We thereby used cryo-EM for direct imaging. To start with, we used the CED-4 apoptosome as a reference. When the buffer for CED-4 crystallization, which contained 150 mM NaCl, was used to prepare cryo-samples, the oligomers were prone to disassemble, leading to a high heterogeneity of the particles (Figs 1B and S1B, blue). When the salt concentration was lowered to 10 mM, the elution peak was shifted to an earlier volume on size exclusion chromatography (SEC) (Fig 1B, red). Consistently, the particles became more homogenous under cryo-condition (Fig S1B, red). We then used the low-salt sample for data collection and followed standard protocols for image acquisition and data processing.

Multiple oligomeric states of the CED-4 apoptosome were observed after global 3D classification. Hexamers, heptamers, and octamers

account for 80.5%, 10.3%, and 9.2%, respectively, of total particles (Fig S2 and Table S2). After 3D refinement, four maps were reconstructed, one for hexamer, one for heptamer, and two for octamers (Fig 1C). For the hexamer, the top ring-like density was of ~ 5.9 Å resolution, and the "broken" funnel was determined to 4.2 Å (Figs S2 and S3 and Table S1). Three CED-4 dimers from the crystal structure (PDB code: 3LQQ) were assigned to the density. The gap of the funnel is not sufficient to accommodate a fourth dimer (Fig 1E). The CARDs from CED-4a (slate) or CED-4b (limon) form the lower and upper CARD ring structure, respectively. These CARDs stack via a previously unobserved mode (Fig 1D and F).

In the crystal structure of the CED-4 apoptosome, the upper CARDs form a ring that is identical to the lower one. The upper ring is rotated by 45 degrees relative to the lower one, leaving each CARD contacted by two (Fig 1F, upper panel). In the reconstructed

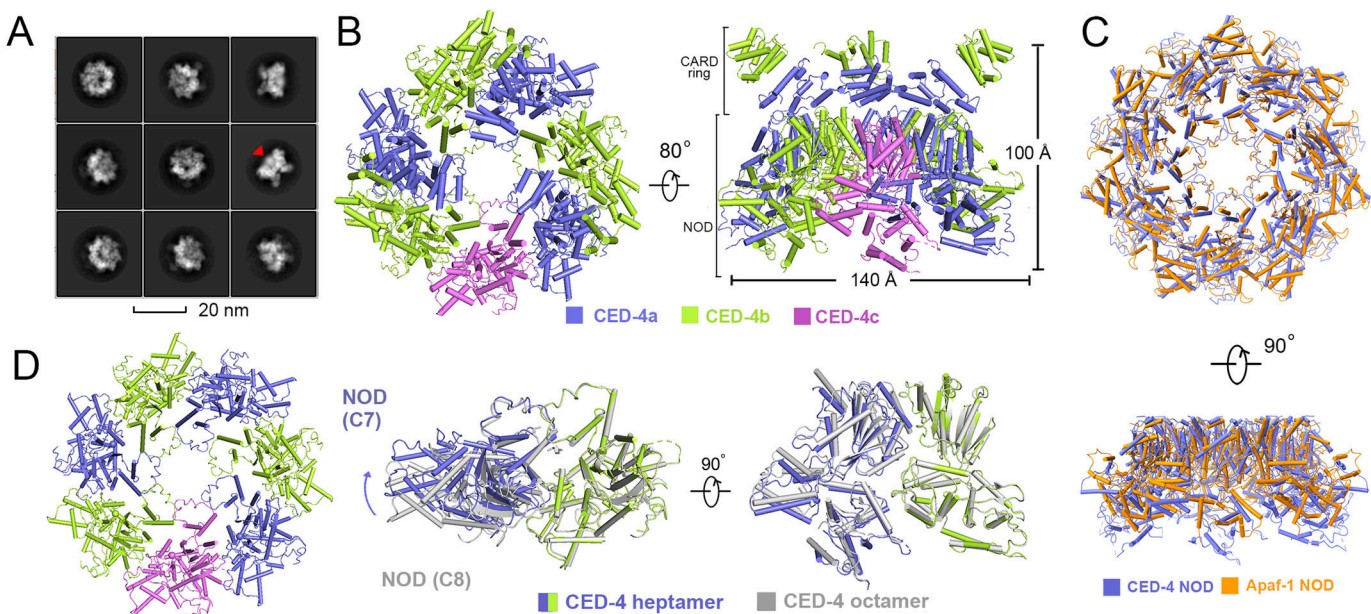

**Figure 2.  Cryo-EM structure of a heptameric CED-4 in the presence of the catalytic domain of CED-3.**
**(A)** Representative 2D class averages of the complex. A globular density in the hutch is indicated by a red arrow. **(B)** Two perpendicular views of the overall structure of a heptameric CED-4/CED-3 catalytic complex. CED-4a and CED-4b are colored slate and limon, respectively. CED-4c, whose CARD is invisible, is colored pink. The catalytic domain of CED-3 cannot be modelled owing to the low resolution. **(C)** The heptameric CED-4 NOD and Apaf-1 NOD are structurally conserved. NODs from the heptameric CED-4 (slate) and the Apaf-1 apoptosome (orange, PDB code: 3JBT) are aligned. **(D)** The oligomerization interfaces of heptameric and octameric (grey) CED-4 NODs are similar.

hexamer, the upper CARDs sit on the periphery of lower CARDs. There is only one interface between each upper and lower CARDs (Fig 1D and F, lower panel). For description simplicity, we name these two stacking modes as group I and group II. In the four maps of the CED-4 apoptosome, only one class of octamer (octamer 2) has group I stacking, whereas group II is seen in hexamers, heptamers, and octamer 1 (Fig 1C). Group II stacking is likely to be maintained by salt bridges between helix H6 and helix H4 of CED-4a and CED-4b (Fig 1F, orange rectangle), in contrast to the multiple interfaces in group I (Figs 1F and S4A) (Qi et al, 2010), explaining its moderate local resolutions (Fig S4B).

### Electron microscopy of the CED-4/CED-3 catalytic complex

The CED-4/CED-3 catalytic complex was prepared by incubating CED-4 with a CED-3 catalytic mutant (198–503, C358A) (Fig S1A, magenta). There were also three oligomeric states after global 3D classification, including 43.1% hexamers, 40.5% heptamers, and 16.4% octamers (Fig S5 and Table S2). In the presence of the CED-3 catalytic domain, the percentage of intact particles, that is, the heptamers and octamers, substantially increased, with the percentage of heptamers increased by about fourfolds. Similar to previous observations, blob of density that likely belongs to CED-3 is seen in the hutch of the heptamer and octamer, but not the broken hexamer.

As the structure of octameric CED-4 was illustrated in detail before (Qi et al, 2010; Huang et al, 2013), here, we focus on the heptamer for analysis. The 3D reconstruction of a heptamer contains a funnel-shaped density, an upper ring-like density and a globular density in the hutch. The funnel was determined to ~ 3.6 Å

resolution by applying a C7 symmetry. The ring-like structure was poorly resolved to three double-layered lobes (Fig S5). After local masking refinement, the resolution was improved to ~ 6.5 Å, supporting rigid-body docking of individual CARD (Fig S3 and Table S1). The globular density in the hutch was clearly seen in both 2D and 3D classifications; however, the resolution could not be improved to support model building despite our various attempts (Figs 2A and S5).

Six CED-4 CARDs, organized via group II stacking, and a C7 symmetric NOD funnel were resolved in the heptamer, with a height of ~100 Å and a diameter of ~140 Å (Fig 2B). The heptameric architecture of NODs in the CED-4 is highly similar to that in the Apaf-1 apoptosome (Fig 2C). Despite different oligomerization numbers, the inter-subunit interface varies little between the heptameric and octameric CED-4 (Fig 2D).

### Cryo-EM analysis of the CED-4/CED-3_CARD complex and the holoenzyme

We then attempted to assemble the holoenzyme. Recombinantly expressed full-length CED-3 was insoluble in several systems (Qi et al, 2010). During activation, the CARD and the catalytic domain of CED-3 are separated after autocleavages. We thereby assembled the ternary complex of CED-4/CED-3_CARD/CED-3 catalytic domain, hereafter referred to as the holoenzyme, to mimic the full-length CED-4/CED-3 complex (Fig S1A, orange). We also examined the CED-4/CED-3_CARD complex as a reference (Fig S1A, green).

In both cases, there were only two oligomeric states observed after 3D classification, hexamers and octamers. Heptamers no longer

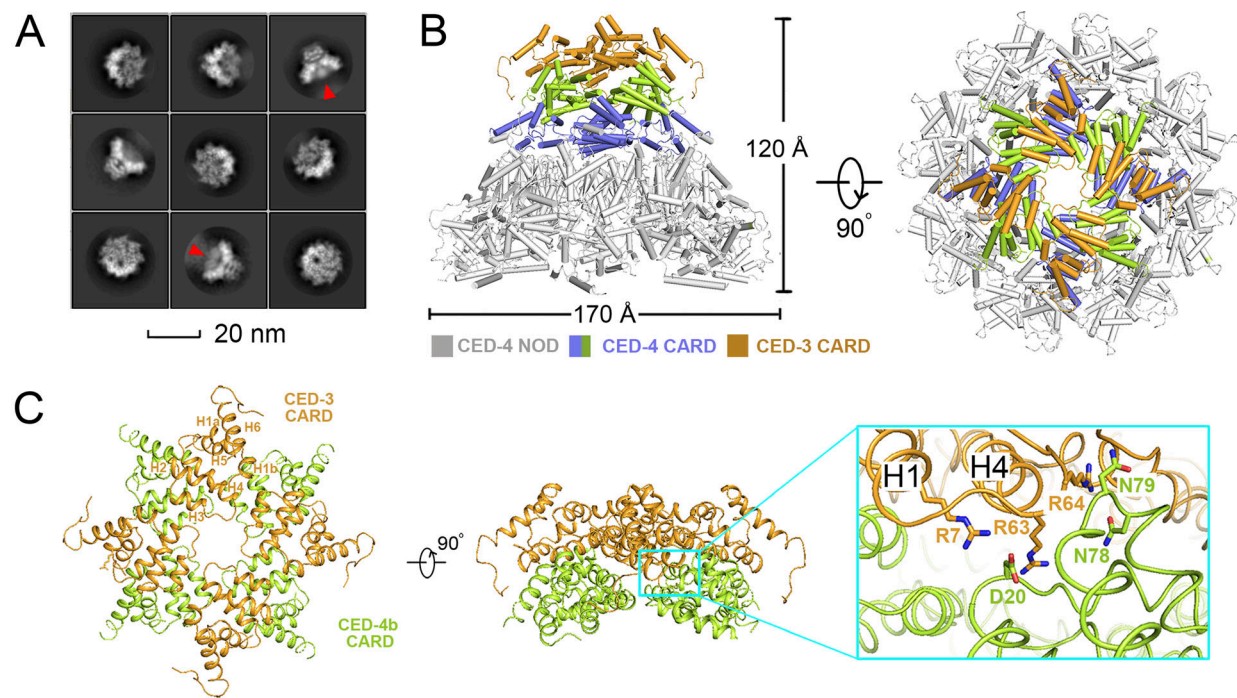

**Figure 3. Cryo-EM structure of an octameric holoenzyme that was assembled by CED-4 and the CARD and catalytic domain of CED-3.**
**(A)** Representative 2D class averages of the holoenzyme. The globular density, which may correspond to the catalytic domain of CED-3, in the hutch is indicated by red arrows. **(B)** Two perpendicular views of the overall structure of the holoenzyme. CED-4 NODs are colored light grey. Upper and lower CED-4 CARD rings are colored limon and slate, respectively. CED-3 CARD ring is in light orange. **(C)** Stacking of the CED-4 CARD and CED-3 CARD rings. The CARD–CARD interface is depicted in detail in the inset. Side chains of putative interacting residues are labeled.

existed in the presence of CED-3 CARD. When treated with CED-3 CARD only, the ratio between hexamer and octamer of CED-4 is 76.7% to 23.3%. In the presence of the catalytic domain of CED-3, more than half of CED-4, 52.1% existed as octamers (Table S2). Other than the density corresponding to the catalytic domain of CED-3 in the hutch, the two CED-3 CARD bound octamers remain nearly identical (Figs 3A, S6, and S7). In the following, we will focus on the holoenzyme for discussion.

## Characterization of the CARD–CARD interface between CED-3 and CED-4

The octameric CED-4 is nearly identical to that in the crystal structure, with the eight CARDs observing a group I stacking. The ring-like density on the top of the CED-4 apoptosome can be assigned with four CED-3 CARDs. Overall, the tower is of ~120 Å tall and ~170 Å wide at the bottom (Fig 3B).

CARD–CARD interaction between CED-3 and CED-4 involves helices H1 and H4 of CED-3. Basic residues on CED-3, including Arg7, Arg63, and Arg64 appeared to contact the acidic residues on CED-4 CARD, including Asp20, Asn78, and Asn79 (Fig 3C, cyan rectangle).

To validate the structural observation, we introduced missense mutations, R7D and R63A/R64A, to CED-3 CARD. Wild type (WT) CED-3 CARD and the two mutants were individually incubated with the CED-4 apoptosome before applying to SEC and SDS–PAGE analysis. The elution peaks of CED-3 mutants were both shifted to later positions. Compared with WT CED-3 CARD, less R7D co-migrated

with CED-4, whereas R63A/R64A no longer associated with CED-4 (Fig 4A). Our biochemical analysis thus corroborated the role of the charged residues in the CARD–CARD interactions.

Taking advantage of these interface-disrupting mutants, we next investigated the impact of the CARD–CARD interaction between CED-4 and CED-3 on CED-3 activation using our previously reported in vitro translation system (Yan et al, 2005). In the absence of the CED-4 apoptosome, CED-3 R7D zymogen activated slowly, similar to WT CED-3. Unexpectedly, CED-3 R63A/R64A showed higher auto-cleavage activity (Fig 4B, lane 1–3). The CED-4 apoptosome evidently accelerated the activation of WT CED-3. In contrast, neither CED-3 R7D nor CED-3 R63A/R64A responded to CED-4 apoptosome, supporting an essential role of the CARD–CARD interaction in CED-3 activation (Fig 4B, lane 4–6).

It is noted that the mammalian homolog Apaf-1 CARD interacts with caspase-9 CARD through two interfaces, of which the type II involves Arg6 and Arg65 of caspase-9 (Hu et al, 2014). In *Drosophila*, CARD–CARD interaction also entails Arg15 and Arg82 of Dronc (Pang et al, 2015). Sequence alignment of the CARDs from caspase-9, Dronc, and CED-3 reveals invariant Arg corresponding to the loci of Arg7 and Arg64 in CED-3 (Fig 4C, red arrows), manifesting the evolutionary conservation of CARD–CARD interactions.

## The CED-4 apoptosome regulates CED-3 activation and activity

As mentioned above, we noticed that auto-activation of CED-3 R63A/R64A was inhibited by CED-4, suggesting an unexpected inhibitory

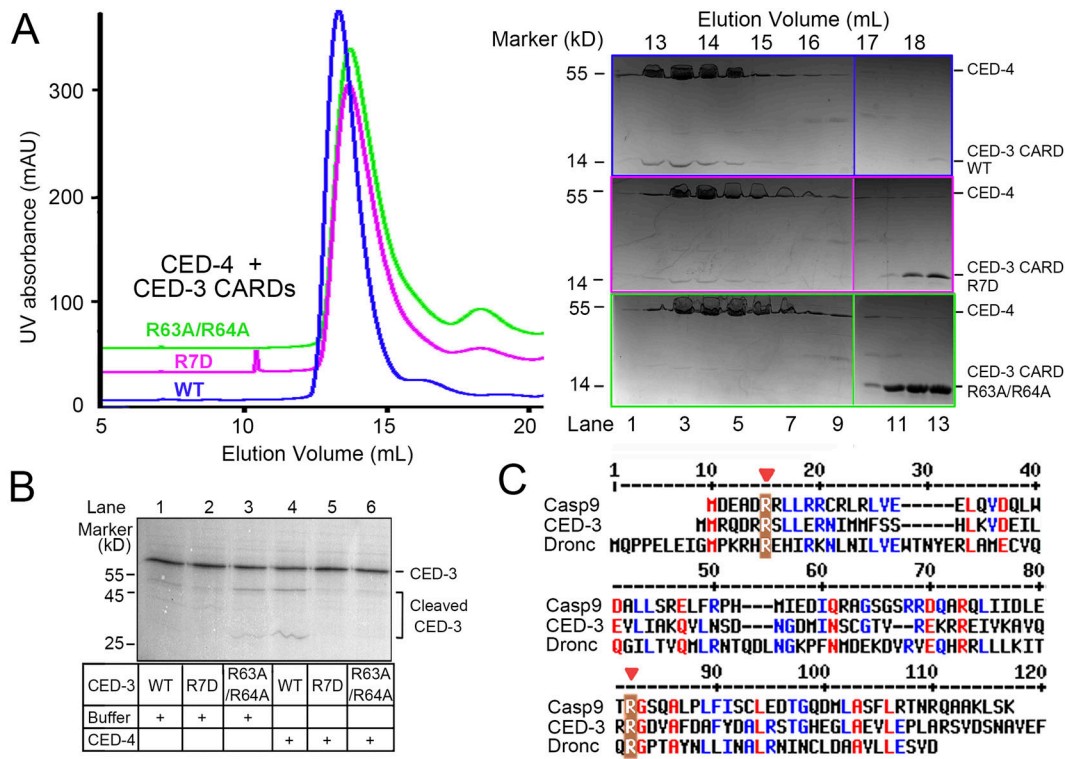

**Figure 4. Biochemical validation of key residues that mediate the CARD–CARD interaction between CED-4 and CED-3.**
**(A)** CED-3 mutations that may compromise the interface disrupted formation of the CED-4/CED-3_CARD complex. Indicated CED-3 CARDs, WT or mutants, were incubated with the CED-4 apoptosome. After SEC, indicated fractions were collected for SDS–PAGE and Coomassie blue staining. **(B)** CARD–CARD interaction is required for CED-3 activation. Biotin-labeled CED-3 variants were in vitro translated for half an hour before CED-4 or buffer was added. After another half an hour, the mixture was applied to SDS–PAGE and visualized through Western blotting against biotin. CED-4 was used at a concentration of 100 nM (monomer). **(C)** Conserved CARD–CARD interface. Shown here is the sequence alignment of CARDs from human caspase-9, CED-3 from *C. elegans*, and Dronc from *Drosophila*. Conserved residues are colored red. Key residues that mediate the CARD–CARD interactions between the initiator caspases and the apoptosomes are shaded brown and pointed by red arrows.

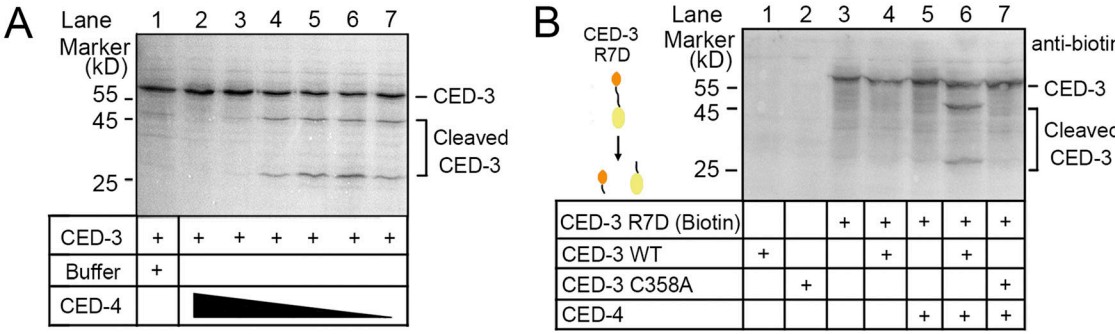

**Figure 5. The CED-4 apoptosome regulates both CED-3 activation and its protease activity.**
**(A)** Dual modulation of CED-3 activation by CED-4 in a concentration-dependent manner. WT CED-3 was generated by in vitro translation. The CED-4 apoptosome used in the assay was 1,000, 250, 125, 62, 31, 16 nM (monomer). **(B)** The CED-4 apoptosome–activated CED-3 acquires enhanced protease activity. Biotin-labeled CED-3 R7D was used as the substrate in the cleavage assay. All CED-3 variants were generated by in vitro translation.

role of the CED-4 apoptosome (Fig 4B, lane 3, 6). We applied a CED-4 gradient to investigate the effect of CED-4 on CED-3 activation. At lower concentrations, CED-4 facilitated CED-3 activation (Fig 5A, lane 4–7). When the concentration was above 250 nM, CED-4 no longer activated CED-3. Moreover, the basal auto-cleavage activity of CED-3 was suppressed by 1 μM CED-4 (Fig 5A, lane 1–3). Therefore, the CED-4 apoptosome modulates CED-3 activation with a dual mechanism.

Then, we investigated the effect of the CED-4 apoptosome on CED-3 protease activity. The enzymatic activity of CED-3 was examined by using the in vitro-translated CED-3 R7D, which cannot be activated by the CED-4 apoptosome, as the substrate. In this assay, only the substrate was biotinylated and thus able to be visualized (Fig 5B, lane 3); WT CED-3 and the catalytic mutant (C358A) were unlabeled (Fig 5B, lane 1, 2). Substrate cleavage was carried out by

incubating enzymes with the substrate for 40 min followed by SDS–PAGE and Western blotting against biotin. As expected, CED-3 zymogens alone had low cleavage activity (Fig 5B, lane 4). CED-4–activated CED-3 efficiently cleaved the substrate (Fig 5B, lane 6), indicating that CED-3 zymogens are substrates of active CED-3 and binding with the CED-4 apoptosome elevates the protease activity of CED-3.

## Discussion

In this study, we report systematic structural and structure-based biochemical analyses that reveal mechanistic insights into CED-3 activation. By using cryo-EM imaging, we were able to capture different oligomeric states of the CED-4 apoptosome, alone or in complex with CED-3. The CED-4 apoptosome, alone or in the presence of the catalytic domain of CED-3, displays three states, including hexamers, heptamers, and octamers. When treated with CED-3 CARD or both the CARD and catalytic domain together (the holoenzyme), only hexamers and octamers of CED-4 were observed. A novel group II stacking of CED-4 CARDs, which shows a high structural flexibility, was found in all the oligomeric states.

By comparing datasets of different CED-4/CED-3 complexes, we conclude that presence of different CED-3 domains tend to stabilize specific oligomeric states of the complexes. CED-3 CARD stabilizes the octameric structure, whereas addition of CED-3 catalytic domain alone stabilizes heptameric assembly (Table S2). Although hexamers existed in all the CED-4/CED-3 complexes, we did not observe similar CED-3 density as seen in the heptamers or octamers.

Then, what is the function of the CED-4 hexamers? In the CED-3 activation assay, we observed an unexpected inhibitory role of the CED-4 apoptosome in CED-3 activation. This phenomenon was not reported in other homologs. It is possible that high concentration of hexamers transiently sequesters a CED-3 monomer, thereby preventing its dimerization or activation. In this scenario, we hypothesize that CED-4 hexamers might represent a pre-mature state of the CED-4 apoptosome and contribute to the regulation of CED-3 activation.

Unlike the mammalian homolog Apaf-1 apoptosome, which binds 4–5 caspase-9 molecules through CARD–CARD interaction (Cheng et al, 2016; Li et al, 2017), the CED-4 apoptosome was reported to bind two molecules of CED-3 catalytic domain in the hutch. However, in the structure of the holoenzyme and the CED-4/CED-3_CARD complex, four CED-3 CARDs were resolved. Then how would the rest two molecules of CED-3 catalytic domain be enzymatically activated? CED-3 zymogens can be cleaved after Asp221, separating the CARD domain and the catalytic domain (Shaham et al, 1999) (Figs 4B and 5). As a result, the rest two molecules of CED-3 catalytic domain can be released and further activated by other CED-4 molecules to contribute to the onset of apoptosis (Qi et al, 2010).

Based on the updated structural and biochemical data, we present a model for CED-3 activation (Fig 6). CED-4 dimers assemble spontaneously after they are relieved from CED-9 sequestration. However, the majority may form pre-mature hexamers. Inactive CED-3 zymogen binds with the CED-4 apoptosome, which initiates CED-3 activation, a process that requires CARD–CARD interaction.

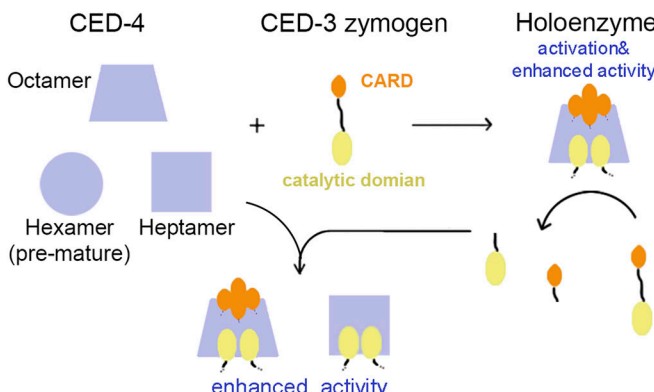

**Figure 6. A working model for the regulation of CED-3 activation and activity.** Most of the CED-4 apoptosome molecules are pre-mature hexamers, which may represent a potential regulatory step. CED-3 activation relies on both CED-4_CARD/CED-3_ CARD and the CED-4/CED-3 catalytic domain interaction. CED-3 zymogens bind to the octameric CED-4, within which it is activated and exhibits enhanced activity. Activated CED-3 cleaves other CED-3 zymogens, separating the CARD domain and the catalytic domain. The released catalytic domain and CARD form a heptameric or octameric complex with CED-4 and the protease activity is elevated.

The activated CED-3 obtains enhanced protease activity to activate more CED-3 zymogens. The CED-3 catalytic domain associates with CED-4 to form heptamer or octamer, in which CED-3 exhibits enhanced protease activities.

Despite advances in cryo-EM imaging, we could not improve the density of the CED-3 catalytic domain in hutch of the CED-4 apoptosome, limiting our understanding of the function of CED-4–associated CED-3. Besides, it is unclear why the double mutation R63A/R64A led to elevated autocleavage activity of CED-3. It remains to be investigated if CED-3 CARD per se plays an inhibitory role in the activation of CED-3.

## Materials and Methods

### Protein purification and complex assembly

The CED-4 apoptosome was obtained as previously reported (Qi et al, 2010). All clones of CED-3 catalytic domain were expressed and purified as described previously (Huang et al, 2013). WT CED-3 CARD and variants were overexpressed with a C-terminal 6x His tag in E.coli BL21 (DE3) cells and purified in 25 mM Tris (pH 8.0), 150 mM NaCl by nickel affinity chromatography (Ni-NTA; QIAGEN), and anion-exchange chromatography (Source-15Q; GE Healthcare). CED-4/CED-3 complexes were assembled at 4°C for 1 h and purified by SEC (Superose-6 [10/30]; GE Healthcare) in 20 mM HEPES (pH 7.5), 10 mM KCl, 1.5 mM MgCl$_2$, 1 mM EDTA, and 1 mM DTT.

### Electron microscopy

3 $\mu$l aliquots of the assembled CED-4 apoptosome or CED-4/CED-3 complexes, at a concentration of ~2 $\mu$M, were applied to glow-discharged Quantifoil 300-mesh Au R1.2/1.3 grids. Grids were blotted in a Vitrobot Mark IV (FEI Company) at 8°C for 2.5 s with 100%

humidity and then plunge-frozen in liquid ethane. Cryo-EM images were recorded automatically on an FEI Titan Krios electron microscope operating at 300 kV. A pixel size of 1.32 Å, defocus values between 1.5 and 2.0 $\mu m$, a dose rate of ~50 e$^-$Å$^{-2}$, and an exposure time of 2.56 s were used on a K2 Summit detector (Gatan Company).

### Image processing

The 32 movie frames of each micrograph were motion corrected by MotionCor2 and binned two fold. Contrast transfer function parameters of the resulting micrographs were estimated by Gctf, and dose weighing was performed. The data processing workflow was similar for the four datasets including the CED-4 apoptosome, the CED-4/CED-3_CARD complex, the CED-4/CED-3 catalytic complex and the holoenzyme. For short, we will use Apo, CARD, catalytic, and holoenzyme to represent the above mentioned four datasets in the following parts, respectively.

Using RELION (version 2.0) (Scheres, 2012), 436,255; 431,187; 599,021; 445,260 particles were auto-picked for Apo, CARD, catalytic, and holoenzyme samples, respectively. After reference-free two-dimensional (2D) classification, 302,443; 271,843; 440,824; 305,373 particles were selected for three-dimensional (3D) classification (Figs S2, S5, S6, and S7). In all cases, the EM map of the CED-4 apoptosome was generated by RELION based on the crystal structure of the CED-4 apoptosome (PDB code 3LQQ) and low-pass filtered to 10 Å as an initial model. Each dataset contained two or three oligomeric states. Apo and catalytic contained hexamers, heptamers, and octamers, whereas CARD and holoenzyme contained hexamers and octamers. 125,140 particles of Apo hexamers, 23,339 particles of CARD octamers, 67,312 particles of catalytic heptamers, and 115,378 particles of holoenzyme octamers were applied for auto-refinement with C1, C4, C7, C4 symmetry, generating 3D reconstructions at overall 4.17 Å, 3.48 Å, 3.56 Å, and 2.99 Å resolution, respectively. For the upper ring-like densities in Apo and catalytic, local masking method was applied and the resolution was improved to 5.93 Å and 6.50 Å resolution. For the upper ring-like densities in CARD and holoenzyme, the method of continuing refinement by adding local mask was applied to improve the map quality, yielding densities of 6.90 Å and 3.80 Å resolution for CARD and holoenzyme, respectively (Fig S3).

### Model building and refinement

The crystal structures of the CED-4 apoptosome (PDB code 3LQQ) and caspase-9 CARD (PDB code 3YGS) were docked into the overall maps by COOT (Emsley & Cowtan, 2004) and fitted into densities by CHIMERA (Pettersen et al, 2004). Initial structural refinement was carried out by PHINEX (Adams et al, 2002).

### Gel filtration analysis of the interaction between CED-4 and CED-3 CARD

The impact of CED-3 CARD mutations on the interaction with the CED-4 apoptosome was assessed by SEC. The CED-4 apoptosome was assembled with WT or mutated CED-3 CARDs at a molar ratio of 1:4 at 4°C for 1 h before loading onto a Superose-6 column (increase 5/150; GE Healthcare). The column was pre-equilibrated with 20 mM HEPES (pH 7.5), 10 mM KCl, 1.5 mM MgCl$_2$, 1 mM EDTA, and 1 mM DTT. Fractions were then examined by SDS–PAGE and Coomassie blue staining.

### In vitro translation assay

In vitro translation of CED-3 zymogen was performed as described (Qi et al, 2010) by using the TnT Quick Coupled Transcription/ Translation Systems (Promega). For CED-3 activation assay, WT and mutated CED-3 zymogens were translated at 30°C for 30 min, then the CED-4 apoptosome or buffer were added to the reaction mixture for another 30 min. For CED-3 activity assay, substrates (CED-3 R7D) were incubated with the activated CED-3 for 40 min. After reaction, samples were mixed with 2× SDS-loading buffer for SDS–PAGE and visualized by the Transcend Non-Radioactive Translation Detection System (Promega).

## Data Availability

The atomic models are available through the PDB with accession codes: 8JNS (CED-4 hexamer), 8JO0 (heptameric CED-4/CED-3 catalytic complex), and 8JOL (octameric holoenzyme). All cryo-EM reconstructions are available through the EMDB with accession codes: EMD-36450 (CED-4 hexamer), EMD-36451 (heptameric CED-4/ CED-3 catalytic complex), and EMD-36459 (octameric holoenzyme).

## Supplementary Information

## Acknowledgements

This work was supported by funds from Beijing Advanced Innovation Center for Structural Biology, Beijing Frontier Research Center for Biological Structure, and Tsinghua-Peking Center for Life Sciences Ministry of Science and Technology. We thank the Tsinghua University Branch of China National Center for Protein Sciences (Beijing) for the cryo-EM facility and the computational facility support. Y Li is supported by fellowships from the Beijing Advanced Innovation Center for Structural Biology and Tsinghua-Peking Center for Life Sciences.

### Author Contributions

Y Li: conceptualization, data curation, formal analysis, validation, investigation, methodology, project administration, and writing—original draft, review, and editing.
L Tian: data curation, formal analysis, visualization, and writing—original draft.
Y Zhang: data curation and formal analysis.
Y Shi: conceptualization, formal analysis, supervision, funding acquisition, project administration, and writing—original draft, review, and editing.

## Conflict of Interest Statement

The authors declare that they have no conflict of interest.

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
