## [Reviewer comments · Life Science Alliance]

Structural insights into CED-3 activation

Yini Li, Lu Tian, Ying Zhang, and Yigong Shi

DOI: <https://10.26508/lsa.202302056>

Corresponding author(s): Yigong Shi, Tsinghua University and Yini Li, Tsinghua University

Review Timeline:

Submission Date:	2023-03-23
Editorial Decision:	2023-04-20
Revision Received:	2023-06-02
Editorial Decision:	2023-06-05
Revision Received:	2023-06-11
Accepted:	2023-06-12

Transaction Report:

April 20, 2023

Re: Life Science Alliance manuscript #LSA-2023-02056-T

Prof. Yigong Shi
Tsinghua University
School of Life Sciences
Biomedical Building
Beijing 100084
China

Dear Dr. Shi,

Thank you for submitting your manuscript entitled "Structural insights into CED-3 activation" to Life Science Alliance. The manuscript was assessed by expert reviewers, whose comments are appended to this letter. We invite you to submit a revised manuscript addressing the Reviewer comments.

Thank you for this interesting contribution to Life Science Alliance. We are looking forward to receiving your revised manuscript.

Sincerely,

B. MANUSCRIPT ORGANIZATION AND FORMATTING:

Reviewer #1 (Comments to the Authors (Required)):

Apoptosis is essential for tissue development and homeostasis. The first knowledge of the apoptotic pathway came from studies of *C. elegans* which revealed the EGL-1/CED-9/CED-4/CED-3 cascade as the central apoptotic pathway in *C. elegans*. Substantial efforts have been made to elucidate the molecular mechanisms of this cascade, however, for a key step of this cascade - activation of CED-3 by CED-4, the detailed mechanism is still not clear. In the current study, Li et al. determined the cryo-EM structures of CED-4 apoptosome alone and in complex with the CARD and catalytic domain of CED-3. Based on these structures, as well as biochemical assays, Li et al. made three important findings: 1) CED-4 apoptosome exists in multiple oligomeric states; 2) Binding of different CED-3 domains stabilizes specific oligomeric states of CED-4 apoptosome; 3) Interactions between CED-3 CARD and CED-4 CARD promotes CED-3 activation. These findings are important pieces of the whole picture of the CED-3 activation mechanism.

Minor comments:

1. In Figure 1 to 3, perpendicular views of the structures are shown. The rotating arrows do not clearly show whether the structures turn 90{degree sign} clockwise or counterclockwise.
2. In Figure 6, the model shows that the CED-3 catalytic domain only associates with CED-4 heptamer, but the CED-3 catalytic domain can also associate with CED-4 octamer. In addition, the model should also include the possibility that both the CARD and the catalytic domain of cleaved CED-3 can bind to the same CED-4 apoptosome.

Reviewer #2 (Comments to the Authors (Required)):

The manuscript by Li, Shi and colleagues describes the cryoEM structures for CED-4 and three different CED-4/CED-3 complexes. Previous crystal structure of CED-4 octamer by the same group reveals a funnel-shaped tetramer of CED-4 dimers, that binds two molecules of mature CED-3. The 8 CARDS of CED-4 form two layers of tetrameric rings. However, the CARD-CARD interaction between CED-4 and CED-3 was not defined due to crystallographic issues. Here the authors prepared three CED-4/CED-3 complexes to investigate CED-4/CED-3 interactions using cryo-EM. The authors observe hexamer, heptamer, and octamer states of CED-4, with increasing percentages of octamers and decreasing hexamers upon addition of CED-3 CARD, catalytic domain, or both. In both heptamers and octamers, densities for CED-3 are observed in the hutch area.

The octamer complex of CED-4/CED-3 shows 4 CARDS from CED-3 pack on top of 8 CARDS from CED-4. Such CARD-CARD interface was validated through CED-4/CED-3 association and CED-4-mediated CED-3 cleavage. Such observation is novel and should be of interest to the readership of LSA.

Two intriguing observations were made as illustrated in figures 4 and 5. First, the auto-activation of CED-3 R63A/R64A mutant was inhibited by CED-4 (please indicate the amount of CED-4 used in figure 4). Second, increasing concentrations of CED-4 above 250 nM not only did not enhance CED-3 activation, it actually suppressed activation. The authors suggest that increasing concentrations of CED-4 may allow some CED-4 to form hexamers, which transiently trap CED-3 in monomeric forms to prevent its activation. Unfortunately, the authors provide no evidence that the hexameric CED-4 is capable of binding CED-3. To support the authors' hypothesis, binding studies of hexameric CED-4 and CED-3 is needed. An alternative hypothesis to the above is that perhaps the octameric CED-4 is capable of binding monomeric CED-3, which effectively reduces CED-3 dimer bound to CED-4 to induce CED-3 auto-cleavage. This may also be tested experimentally.

It is unfortunate that no information was discerned regarding the density or CED-3 domains supposedly bound at the "hutch" of CED-4, which may reveal mechanisms of CED-3 auto-cleavage, and resolve the potential inconsistency of whether 4 CED-3 (this study) or 2 CED-3 (the Qi 2010 paper) bind to CED-4.

Minor issue:

The manuscript uses type I and type II stacking to refer to the organization of the CED-4/CED-3 complexes that involve 2-3 CARDS. This can be confused with the type I, II, and III interfaces between two death domain family members that have been well established in the field. Suggest use category I/II or group I/II stacking.

Reviewer #3 (Comments to the Authors (Required)):

In this paper, Li et al. report cryo-EM structures of CED-4 and CED-4/CED-3 complexes which represent different activation states. It was observed that CED-4 (alone or in complex with CED-3) is capable of forming hexamers, heptamers and octamers, respectively. Apoptosis play an essential role in the development of multi-cellular organism. The work reveals how the CED-3 and CED-4 apoptosomes specifically interact to recruit caspases. Therefore, the topic of this paper is significant. The work is of high quality. Overall, this is an excellent paper that makes significant contributions to understanding the mechanism of CED-3 activation.

Minor comments:

On p.3 (the last sentence of the 1st paragraph of Introduction), please provide the full name of CED. What does CED stand for?

On p.6 (1st paragraph), it was stated "The two rings rotate by 45 degrees, ...". Probably, the cryo-EM data would not allow for the observation of the process of this rotation? The authors may want to rephrase this sentence.

On p.12 (last paragraph), the authors stated "Despite the advancement in understanding the activation of CED-3 by CED-4, we could not improve the density of CED-3 ..." It doesn't seem like "despite the advancement in understanding the activation" and "we could not improve the density" are related?

In Figs. 1, 2 and 3, the secondary structural elements should be labeled.

In Fig. 6, is it a better idea to use hexagon, heptagon and octagon to represent hexamer, heptamer and octamer (but this is very minor)?

Reviewer #1:

This reviewer is positive about our research. They only raised minor concerns that are addressed below:

1. *In Figure 1 to 3, perpendicular views of the structures are shown. The rotating arrows do not clearly show whether the structures turn 90{degree sign} clockwise or counterclockwise.*

Point taken. Figures with corrected arrows are provided.

2. *In Figure 6, the model shows that the CED-3 catalytic domain only associates with CED-4 heptamer, but the CED-3 catalytic domain can also associate with CED-4 octamer. In addition, the model should also include the possibility that both the CARD and the catalytic domain of cleaved CED-3 can bind to the same CED-4 apoptosome.*

CED-3 catalytic domain can associate with CED-4 octamer based on our cryo-EM data of the CED-4/CED-3 catalytic complex, but considering it is a minor group and densities for CED-3 catalytic domain are weak, we did not show it in our model. It is possible that both the CARD and the catalytic domain can bind to the same CED-4 apoptosome, so we add this active holoenzyme in our model.

We thank this reviewer for their constructive comments.

Reviewer #2:

This reviewer is generally positive about our research. They raised several concerns that are addressed below:

1. *First, the auto-activation of CED-3 R63A/R64A mutant was inhibited by CED-4 (please indicate the amount of CED-4 used in figure 4).*

Point taken. The concentration of CED-4 used in the activation assay was added in the corresponding figure legend.

2. *Unfortunately, the authors provide no evidence that the hexameric CED-4 is capable of binding CED-3. To support the authors' hypothesis, binding studies of hexameric CED-4 and CED-3 is needed. An alternative hypothesis to the above is that perhaps the octameric CED-4 is capable of binding monomeric CED-3, which effectively reduces CED-3 dimer bound to CED-4 to induce CED-3 auto-cleavage.*

This may also be tested experimentally.

We appreciate this insightful comment on the mechanism of CED-4 inhibition. Actually, we did observe an extra density in the hexameric CED-4 and it was off-center (Fig. 1), different from the globular density observed in the hutch of a CED-4 octamer. However, the density was weak, indicating that the binding is unstable even at high protein concentrations. Current resolution was not sufficient for us to draw any conclusion, so we just brought out one possibility that "CED-3 hexamers transiently trap CED-3 in monomeric forms to prevent its activation". In order to perform binding assay, we tried to fractionalize CED-4 hexamers, heptamers and octamers, but unfortunately, we failed possibly due to their high flexibility and similar molecular weights.

[Figure removed by editorial staff per authors' request]

3. *It is unfortunate that no information was discerned regarding the density or CED-3 domains supposedly bound at the "hutch" of CED-4, which may reveal mechanisms of CED-3 auto-cleavage, and resolve the potential inconsistency of*

whether 4 CED-3 (this study) or 2 CED-3 (the Qi 2010 paper) bind to CED-4.

In the manuscript, we reported 4 CED-3 CARDS in the holoenzyme, and Qi et al. reported 2 CED-3 catalytic domains of the CED-3 in CED-4 hutch. We agree with Qi's paper because CED-4 hutch is only big enough to accommodate 2 CED-3 catalytic domains. We discussed the mismatch of the number of bound CED-3 domains in the Discussion session. We found that CED-3 zymogens can be cleaved by active CED-3 (see Figure 5 in the manuscript), so for CED-3 that binds to CED-4 only through its CARD, its catalytic domain can be released and then binds to other CED-4 molecules.

Minor issue:

The manuscript uses type I and type II stacking to refer to the organization of the CED-4/CED-3 complexes that involve 2-3 CARDS. This can be confused with the type I, II, and III interfaces between two death domain family members that have been well established in the field. Suggest use category I/II or group I/II stacking.

Point taken. The terms were replaced with group I and group II.

We thank this reviewer for their constructive comments.

Reviewer #3:

This reviewer thinks highly of our work. A few minor comments are addressed below.

1. *On p.3 (the last sentence of the 1st paragraph of Introduction), please provide the full name of CED. What does CED stand for?*

We add the full name of CED on Page.3.

2. *On p.6 (1st paragraph), it was stated "The two rings rotate by 45 degrees, ...". Probably, the cryo-EM data would not allow for the observation of the process of this rotation? The authors may want to rephrase this sentence.*

On Page.6, the statement of "The two rings rotate by 45 degrees, ..." might be confusing. We rephrase it as "In the crystal structure of the CED-4 apoptosome, the upper CARDS form a ring that is identical to the lower one. The upper ring is rotated by 45 degrees relative to the lower one, leaving ...".

3. *On p.12 (last paragraph), the authors stated "Despite the advancement in understanding the activation of CED-3 by CED-4, we could not improve the density of CED-3 ..." It doesn't seem like "despite the advancement in understanding the activation" and "we could not improve the density" are related?*

On Page.12, we rephrase the sentence as "Despite advances in cryo-EM imaging, we could not improve the density of the CED-3 catalytic domain ...".

4. *In Figs. 1, 2 and 3, the secondary structural elements should be labeled.*

Thanks for your suggestion in labeling. We add secondary structure elements of CED-3 CARD in Figure 3C. In Figure 1-3, models of CED-4 hexamers, heptamers and octamers were built by docking CED-4 molecules from the published crystal structure of the CED-4 apoptosome. Besides, we focused on different oligomeric states of CED-4 in these figures. Therefore, we did not label secondary structure elements or discuss them in our manuscript.

5. *In Fig. 6, is it a better idea to use hexagon, heptagon and octagon to represent hexamer, heptamer and octamer (but this is very minor)?*

Thank you for your advice in depicting the working model. We had considered to use polygons to represent different oligomeric states. Heptagon and octagon can well represent our structures of heptamers and octamers, but the hexamer we observed is an opened structure and it resembles to an octamer that misses a CED-4 dimer. We think that a hexagon might be misleading and confusing, therefore we did not use polygons in our model.

We thank this reviewer for their constructive comments.

June 5, 2023

RE: Life Science Alliance Manuscript #LSA-2023-02056-TR

Prof. Yigong Shi
Tsinghua University
School of Life Sciences
Biomedical Building
Beijing 100084
China

Dear Dr. Shi,

Thank you for submitting your revised manuscript entitled "Structural insights into CED-3 activation". We would be happy to publish your paper in Life Science Alliance pending final revisions necessary to meet our formatting guidelines.

- please add ORCID ID for secondary corresponding author--they should have received instructions on how to do so
- please add the Twitter handle of your host institute/organization as well as your own or/and one of the authors in our system
- please add your main, supplementary figure, and table legends to the main manuscript text after the references section;
- please consult our manuscript preparation guidelines <https://www.life-science-alliance.org/manuscript-prep> and make sure your manuscript sections are in the correct order (The introduction section is not labeled, etc...)
- please add an Author Contributions section to your main manuscript text
- please add a conflict of interest statement to your main manuscript text
- please add a callout for Figures 3A and S1A to your main manuscript text
- please make sure to reference Table S1 in the manuscript text properly

A. FINAL FILES:

B. MANUSCRIPT ORGANIZATION AND FORMATTING:

Sincerely,

June 12, 2023

RE: Life Science Alliance Manuscript #LSA-2023-02056-TRR

Prof. Yigong Shi
Tsinghua University
School of Life Sciences
Biomedical Building
Beijing 100084
China

Dear Dr. Shi,

Thank you for submitting your Research Article entitled "Structural insights into CED-3 activation". It is a pleasure to let you know that your manuscript is now accepted for publication in Life Science Alliance. Congratulations on this interesting work.

DISTRIBUTION OF MATERIALS:

Again, congratulations on a very nice paper. I hope you found the review process to be constructive and are pleased with how the manuscript was handled editorially. We look forward to future exciting submissions from your lab.

Sincerely,
